# The age profile of respiratory syncytial virus burden in preschool children of low- and middle-income countries: A semi-parametric, meta-regression approach

**Marina Antillón**[1,2,3]*, **Xiao Li**[1], **Lander Willem**[1], **Joke Bilcke**[1], **RESCEU investigators, Mark Jit**[4], **Philippe Beutels**[1]

**1** Center for Health Economics and Modeling of Infectious Diseases, University of Antwerp, Antwerp, Belgium, **2** Swiss Tropical and Public Health Institute, Allschwil, Switzerland, **3** University of Basel, Basel, Switzerland, **4** London School of Hygiene and Tropical Medicine, London, United Kingdom

* marina.antillon@aya.yale.edu

## Abstract

**Data Availability Statement:** All coding files are available from the following github repository https://github.com/Marina-Antillon/rsv_splines_lmics.

### Background

Respiratory syncytial virus (RSV) infections are among the primary causes of death for children under 5 years of age worldwide. A notable challenge with many of the upcoming prophylactic interventions against RSV is their short duration of protection, making the age profile of key interest to the design of prevention strategies.

### Methods and findings

We leverage the RSV data collected on cases, hospitalizations, and deaths in a systematic review in combination with flexible generalized additive mixed models (GAMMs) to characterize the age burden of RSV incidence, hospitalization, and hospital-based case fatality rate (hCFR). Due to the flexible nature of GAMMs, we estimate the peak, median, and mean incidence of infection to inform discussions on the ideal "window of protection" of prophylactic interventions. In a secondary analysis, we reestimate the burden of RSV in all low- and middle-income countries. The peak age of community-based incidence is 4.8 months, and the mean and median age of infection is 18.9 and 14.7 months, respectively. Estimating the age profile using the incidence coming from hospital-based studies yields a slightly younger age profile, in which the peak age of infection is 2.6 months and the mean and median age of infection are 15.8 and 11.6 months, respectively. More severe outcomes, such as hospitalization and in-hospital death have a younger age profile. Children under 6 months of age constitute 10% of the population under 5 years of age but bear 20% to 29% of cases, 28% to 39% of hospitalizations, and 38% to 50% of deaths.

On an average year, we estimate 28.23 to 31.34 million cases of RSV, between 2.95 to 3.35 million hospitalizations, and 16,835 to 19,909 in-hospital deaths in low, lower- and upper middle-income countries. In addition, we estimate 17,254 to 23,875 deaths in the community, for a total of 34,114 to 46,485 deaths. Globally, evidence shows that

**Funding:** Respiratory Syncytial Virus Consortium in Europe (RESCEU) has received funding from the Innovative Medicines Initiative 2 Joint Undertaking under grant agreement No 116019 (XL, LW, JB, MJ, PB). which receives support from the European Union's Horizon 2020 research and innovation programme and the European Federation of Pharmaceutical Industries and Associations (EFPIA). MA has received funding from a post-doctoral fellowship of the Belgian-American Education Foundation. PB also received support through PATH by Gavi, the Vaccine Alliance. The funders of the study had no role in study design, data collection, data analysis, data interpretation, or writing of the report.

**Competing interests:** I have read the journal's policy and the authors of this manuscript have the following competing interests: LW received grants from Research Foundation Flanders (FWO) during the conduct of the study and fees from Pfizer outside the submitted work for discussions on economic evaluation, for a total of <€3000 combined, fully paid directly to the University of Antwerp. PB reports a grant from the Respiratory Syncytial Virus Consortium in Europe (RESCEU), Innovative Medicines Initiative 2 of the European Commission, Joint Undertaking under grant agreement No 116019, during the conduct of the study; and outside the submitted work he reports grants from Pfizer, GSK, Merck and the Innovative Medicines Initiative 2 of the European Commission (N° 101034339 project PROMISE: Preparing for RSV immunisation and surveillance in Europe) as well as consultancy fees.

**Abbreviations:** ALRI, acute lower respiratory infection; BM, burden model; CFR, case fatality rate; DF, degrees of freedom; FE, fixed-effects; GAMM, generalized additive mixed model; GLMM, generalized linear mixed model; hCFR, hospital-based case fatality rate; LIC, low-income country; LMIC, lower-middle-income country; OM, outcomes model; RE, random-effects; RSV, respiratory syncytial virus; UMIC, upper-middle-income country.

community-based incidence may differ by World Bank Income Group, but not hospital-based incidence, probability of hospitalization, or the probability of in-hospital death ($p \leq 0.01$, $p = 1$, $p = 0.86$, $0.63$, respectively). Our study is limited mainly due to the sparsity of the data, especially for low-income countries (LICs). The lack of information for some populations makes detecting heterogeneity between income groups difficult, and differences in access to care may impact the reported burden.

## Conclusions

We have demonstrated an approach to synthesize information on RSV outcomes in a statistically principled manner, and we estimate that the age profile of RSV burden depends on whether information on incidence is collected in hospitals or in the community. Our results suggest that the ideal prophylactic strategy may require multiple products to avert the risk among preschool children.

## Author summary

### Why was this study done?

- Respiratory syncytial virus (RSV) is the most common cause of acute pulmonary infections in children. The RSV disease burden is high, especially in the nearly 600 million children under 5 living in 121 low-income (LIC) and middle-income countries (MICs) on which this study focuses.

- Evidence on the age distribution of RSV infections in these countries is based on sparse data using age breakdowns that are not always comparable.

- Different pharmaceutical products are becoming available that can reduce the RSV burden. Given that the immunity these products confer differs and wanes over time, it is essential to understand well at which months of age RSV infections drive the RSV disease burden.

- This study uses improved statistical models to estimate in depth the age profile of RSV cases, hospitalizations, and in-hospital deaths in young children.

### What did the researchers do and find?

- We calculated the distributions of the age of infection, hospitalization, and in-hospital deaths. Depending on whether we use hospital-based or community-based incidence studies to inform our methods, we estimate the peak age of infection at 2.6 to 4.8 months, the mean age at 15.8 to 18.9, and the median age at 11.6 to 14.7 months.

- We estimate that on an average year, there are 28.23 to 31.34 million cases of RSV, 2.95 to 3.35 million hospitalizations, and 34,114 to 46,485 deaths in children under 5 in LICs and MICs. About half the deaths occur in the community, outside of hospital settings.

- More severe outcomes, such as hospitalizations and in-hospital deaths have a younger age profile. Children under 6 months of age constitute 10% of the population under 5

years of age but bear 20% to 29% of cases, 28% to 39% of hospitalizations, and 38% to 50% of deaths.

### What do these findings mean?

- Our results support strategies using passive immunity products, such as maternal vaccines and monoclonal antibodies, to protect infants and active vaccination strategies for children over one, who also bear a large proportion of the burden.

- These results improve the choice of strategies offering the best value for money from a given budget.

- This study may enable modelers to make improved estimates thus allowing policy-makers to gain a better understanding of the potential impact that new pharmaceutical products could have.

## Introduction

Respiratory syncytial virus (RSV) infections are responsible for the largest proportion of acute lower respiratory infections (ALRIs) in low- and middle-income countries, and by extension, it is among the top 5 causes of death in children under 5 years of age [1,2]. Among severe ALRIs in children under 5 years of age in low-income countries (LICs), lower-middle-income countries (LMICs), and upper-middle-income countries (UMICs), RSV is the single largest cause of pneumonia, accounting for 31% of cases [2]. RSV transmission is determined by various factors, maternally conferred immunity [3–5], social contact patterns [6–10], and the short duration of immunity from infection [11–13].

Epidemiological questions surrounding RSV in early life are now more pressing because of (1) ongoing research and development of multiple prophylactic products—ranging from vaccines administered to mothers and children to monoclonal antibodies administered to infants [14–20]; and (2) ongoing policy discussions for optimal use of prophylaxis in resource-constrained settings [21–26]. Maternally derived antibodies of RSV have a half-life of 36 to 38 days [4,5]. Chu and Englund's systematic review found maternal vaccines—against any disease—protect children no longer than 4 months [27]. A currently licensed short-acting monoclonal antibody, palivizumab, requires monthly administrations [16,17]. Newer antibodies with "extended" half-lives last between 62 to 150 days [18,19]. Recently announced topline results for Pfizer's ongoing trial of a RSVpreF maternal vaccine showed 69% reduction in medically attended lower-respiratory tract infections through 6 months of age. An additional set of products under development are active immunizations or childhood vaccines, which could work as a follow-up product to maternal vaccines or monoclonal antibodies administered in the first few weeks of life [20,28,29]. In light of the importance of the duration of protection and the presence of a potential high cost, it is imperative to develop detailed age-specific estimates of incidence, hospitalization, and mortality that can be incorporated with careful economic analyses that weigh the costs and the benefits of options for using these products.

The current evidence on the age distribution of RSV infections in low- and middle-income countries is based on sparse data. While the systematic review by Shi and colleagues and updated by Li and colleagues found data from numerous (previously unpublished) hospital-based studies, the age profile of RSV incidence remains elusive because studies present the

data in broad differing age bands, making comparison and synthesis a challenge [30,31]. Among studies that presented sufficient data to evaluate the age profile of RSV burden (data stratified in 3 or more age groups), there were only 18 community-based incidence studies in low and middle-income countries and, of these, only 8 studies presented the probability of hospitalization among cases in the community. Therefore, although there are numerous studies on the age profile of hospitalized cases in many settings, the age-specific relationship between community and hospital incidence remains unclear.

Moreover, the epidemiological data as it is presented in the literature is often discretised using 1 to 7 age groups under the age of 5 years [30,31] rendering the incidence across early childhood as only a partially observed process which could be interpolated by employing a strong assumption of monotonicity [30–34]. The interpolation of RSV incidence, hospitalization, and deaths throughout early childhood for different settings requires potentially influential assumptions with consequences for health policy [21].

An alternative approach has been carried out to inform vaccine interventions. Rather than estimating absolute incidence, one study based in Kenya estimated the ideal "vaccine window" using a combination of serological data and nested catalytic models to estimate the optimal age for prophylaxis [35]. However, the vaccine window has not been estimated across different settings or reexamined using community and clinical outcome data.

In the current study, we leverage the data collected by Shi and colleagues and Li and colleagues in combination with an alternative statistical approach to present estimates that circumvent the shortcomings listed above. We use random-effects generalized additive models, a family of semi-parametric models to construct nonlinear and non-monotonic piece-wise defined functions that characterize the trends of RSV across early childhood. By characterizing the age distribution of RSV incidence, hospitalization, and death among young children across different settings, we reestimate the peak, median, and mean age of infection, therefore, informing discussions on the ideal "vaccine window" of a prophylactic drug with a defined duration of protection. In a secondary analysis, we reestimate and reconsider the burden of RSV for different countries and country groups.

## Methods

### Data

Data on RSV cases in the community, hospitals, and in-hospital deaths were extracted from Shi and colleagues' and Li and colleagues' systematic reviews, restricting to studies carried out with an end date in 2000 or later, as this reflects more recent disease patterns, diagnostic practices, and treatment success rates [30,31]. Table 1 shows an overview of the studies available by country income group and outcome of interest—community- and hospital-based incidence, the probability of hospitalization, and the probability of death. We used studies that reported age-specific incidence in at least 3 mutually exclusive age groups for the estimation and studies that used fewer age groups for out-of-sample validation. Modifications to the data provided by Shi and colleagues and Li and colleagues for the purpose of computational compatibility with our approach are detailed in Section S1.1 in S1 Text. These modifications are primarily to distill the data to mutually exclusive age groups and to transform reported incidence and confidence intervals to counts of cases and denominator (population-time).

### Model of RSV epidemiology

Since the data on RSV epidemiology in infants and preschool children is usually presented in age groups spanning several months or up to a year, making the month-by-month age-specific epidemiology a partially observed process, we sought a method to estimate monthly

**Table 1. Number of studies available for each RSV-related outcome, in total and broken down by World Bank income category.**

| | All countries | Low income | Low-middle income | Upper-middle income |
|---|---|---|---|---|
| **Spline I: Community-based incidence** | | | | |
| Estimation dataset | 18 (S:15, VS:4) | 0 | 12 (S:11, VS:4) | 6 (S:4) |
| Validation dataset | 2 | 0 | 2 | 0 |
| **Spline II: Hospital-based incidence** | | | | |
| Estimation dataset | 37 (S:15, VS:13) | 5 (S:2, VS:3) | 14 (S:5, VS:4) | 18 (S:8, VS:6) |
| Validation dataset | 15 (S: 1, VS: 1) | 2 | 8 (S: 1, VS: 1) | 5 |
| **Spline III: Probability of hospitalization among cases in the community** | | | | |
| Estimation dataset | 8 | 0 | 5 | 3 |
| Validation dataset | 3 | 0 | 2 | 1 |
| **Spline IV: Probability of death among hospitalized cases** | | | | |
| Estimation dataset | 58 | 9 | 24 | 25 |
| Validation dataset | 30 | 3 | 8 | 19 |

For outcomes I and II, the numbers in parentheses indicate the number of studies that have additional information regarding severe (S) and very severe cases (VS). Studies where the outcome was stratified for at least 3 age groups were used to estimate the splines, and studies for which the outcome was presented for only 1 or 2 age groups were used as a form of out-of-sample validation. For a visual of the geographic distribution of different data types, see Fig A in S1 Text.

interpolations of incidence, hospitalization, and mortality before the age of 5. However, not all of the measures of interest are available from all settings where studies were run, resulting in a fragmented understanding of how cases, hospitalizations, and deaths are related. Therefore, we adopted an approach that leverages the age-specific data on each of these outcomes where available, integrating the extant age-specific data on RSV epidemiology into a burden model (BM) that assumes that community, hospitalized, and fatal cases in hospitals follow a multiplicative relationship, as shown in Fig 1. In other words, within this framework, the probability that a case in the community becomes a case in the hospital and that the hospitalized case becomes a fatality is itself age dependent. Moreover, we compare 2 outcomes models (OMs): OM I, in which the starting point is community-based incidence, and then the conditional probability of hospitalization given infection and in-hospital death are applied, and the second, OM II, in which the starting point is hospital-based incidence, and the conditional probability of hospitalization is used to back-calculate community incidence and the probability of death is applied to calculate fatalities. In the absence of any reason to choose one model over another, we look at the results of both side-by-side.

Because RSV outcomes do not necessarily have linear or monotonous relationships with age, we estimated interpolation splines within the generalized additive mixed model (GAMM) framework, a semi-parametric approach [36,37]. Unlike fully parametric models in which the predictors and the outcomes are assumed to have a linear or a linear-transformed relationship—potentially a strong or unwarranted assumption—GAMMs sum polynomials to produce an interpolation spline, yielding a highly flexible modeling framework. Four predictive functions were estimated:

- Spline I: age-specific incidence as measured in community-based studies.

- Spline II: age-specific incidence as measured in hospital-based studies.

- Spline III: age-specific conditional probability of hospitalization given (measured in community-based studies).

- Spline IV: age-specific conditional probability of death given hospitalization (hCFR) (measured in hospital-based studies).

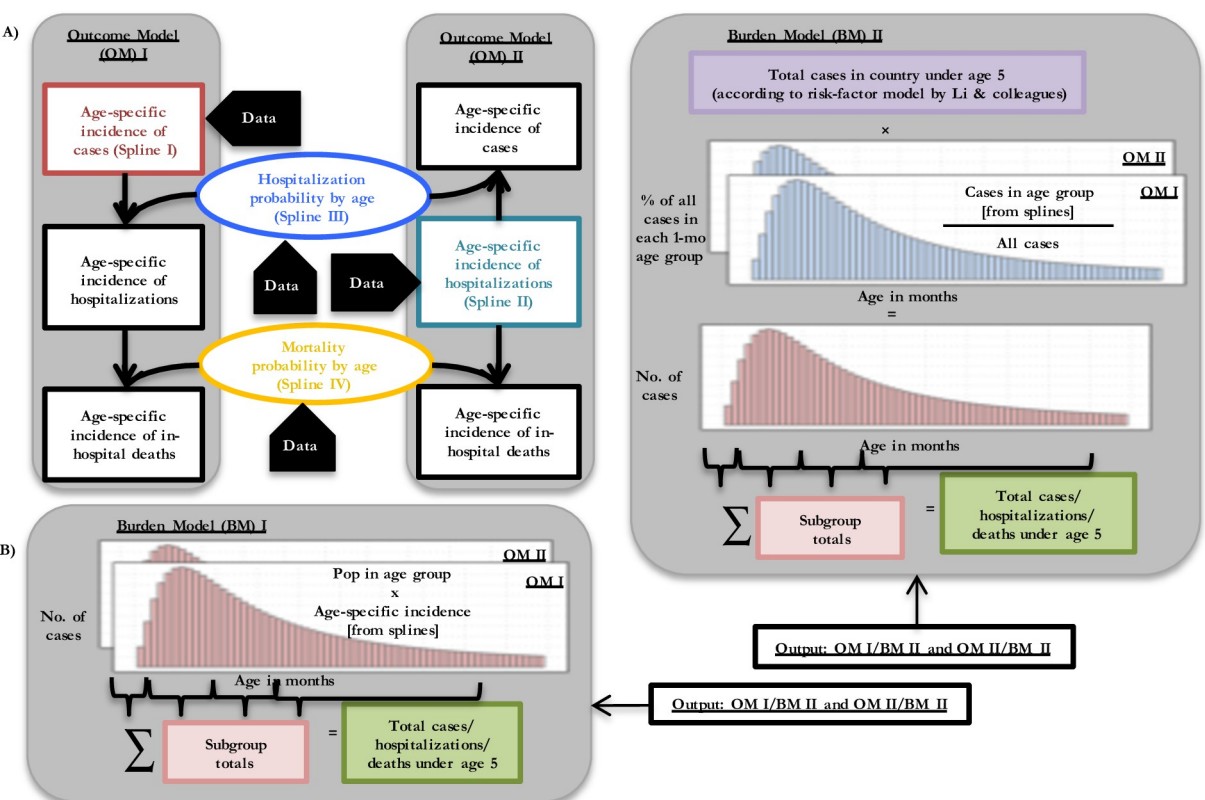

**Fig 1. Relationship between cases, hospitalizations, and deaths in children from birth to 59 months of age. (A) OMs (age-specific).** There are 2 ways to construct outcomes splines of the number of cases, hospitalizations, and deaths from the data in the literature, corresponding to OM I and II. The boxes show incidence and the ovals show the probability of progressing from a case in the community to a case in the hospital or from a case in the hospital to a fatal case. It should be noted that the product of incidence and probability yields an incidence. The colored boxes show the splines (splines I–IV) that we estimated from data in the literature, and the black boxes show incidence derived as products of our splines. In OM I, we use the incidence of community-acquired RSV cases (spline I) and the probability of hospitalization given infection (spline III) and deaths among inpatients (spline IV) and we derive the incidence of hospitalizations and the incidence of death due to RSV. In OM II, we use the incidence of RSV cases from hospital-based studies (spline II) and the probabilities of hospitalization and inpatient death and we back-calculate the community-based incidence and the incidence of death due to RSV. From these spline models, we also estimated the peak, median, and mean age of infection. **(B) BMs of disease in population.** There are 2 ways to calculate the burden of disease in one country: for BM I, we apply the country population size to the incidences derived in (A) and aggregate cases into subgroups according to age; for BM II, we take the number of cases in the country as calculated by Li and colleagues' risk-factor model, apply the proportion of cases that occur in each month of age according to our splines, and aggregate cases into subgroups according to age. Because there are 2 ways to calculate age-specific incidence in part (A) and 2 ways to calculate burden in part (B), 4 sets of burden estimates result. A full mathematical derivation of this is found in Section S1.4 in S1 text. BM, burden model; OM, outcomes model; RSV, respiratory syncytial virus.

Furthermore, we tested the significance of the income group (as designated by the World Bank in 2020) as a modifier of the age-dependent spline. Our baseline model was one that included a single (global) spline to explain the relationship between the outcome and age. Then, we tested whether a fixed-effect spline stratified by income group was enough to capture the differences between countries or whether both a fixed-effect and study-specific random-effect splines stratified by income group were necessary to capture the trends. Models were compared using the generalized likelihood ratio test using the $\chi^2$ statistic. Further details are found in Section S1.4 in S1 Text.

We projected each of the splines by sampling 5,000 times from a multivariate normal distribution described by the model coefficients to the link function (log for incidence, logit for probability) and back-transformed to present incidence or probabilities as appropriate. This allowed us to propagate the uncertainty of each of the splines throughout the outcome and

burden models described below. Further details on our estimates of uncertainty are found in Sections S.1.5.1 to S1.5.2 in S1 Text.

**Outcome models: Cases, hospitalizations, and deaths among hospitalizations.**   The information from the literature can be used in 2 different ways to calculate the number of cases, hospitalizations, and deaths, corresponding to outcome models (OMs) I and II (Fig 1). In OM I, we use the age-specific incidence from community-acquired RSV cases and the age-specific probabilities of hospitalization and death to derive the incidence of hospitalizations and in-hospital death due to RSV, respectively. In OM II, we back-calculate community-based cases in each age group by dividing the incidence of hospitalized cases by the age-specific probability of hospitalization, and we calculate the number of in-hospital deaths in each age group among hospitalized cases using the incidence from hospital-based studies and the probability of death in hospitalized cases. For further details on our construction of the splines, see Section S1.4 in S1 Text.

**Age profile of RSV burden.**   To inform prophylactic interventions against RSV, to provide policy-relevant summaries of the key ages of the burden of RSV disease, and hence to define the ideal "window of protection", we chose to present 3 measures of the full age distribution of disease: the mean age of infection (often presented in studies), the median age of infection (when half of the cases under 5 have already occurred), and the peak age of infection (the age in which incidence, hospitalizations, and deaths were highest). These 3 measures were calculated for each sample projection of the predictive model of the splines. The accumulated mean, median, and peak from each sample then allowed us to calculate credible intervals for the mean, median, and peak age of infection for the 3 epidemiological outcomes of interest: RSV incidence, hospitalizations, and hospital-based deaths. Further details on our calculations are found in Section S1.7 in S1 Text.

**Burden models: Global model and country-specific model.**   The burden of disease can be estimated using 2 approaches. In BM I, we apply the country population size to the derived incidence as described in Section S1.6 in S1 Text and aggregate cases into subgroups according to age. For BM II, we take the number of cases in children under 5 in each country as calculated by Li's risk-factor model, and we disaggregate that total by applying the proportion of cases that occur in each age group according to our splines (see Fig 1 for a heuristic illustration and Section S1.6 in S1 Text for a mathematical derivation of the approach).

We calculate cases, hospitalizations, and incidence of deaths—in the hospital and in the community—for 121 countries, which are the number of LIC, LMIC, and UMIC countries in Li's risk-factor model. Of those countries, 27 were LICs (GNI per capita ≤$1,045) in 2020 with 103 M children under 5, 53 were LMICs (GNI per capita $1,046 to $4,095) with 335 M children under 5, and 41 were UMICs (GNI per capita $4,096 to $12,695) with 161 M under 5, for a total of 598 M children under 5. For more information on the World Bank country income group and included countries, see Section S1.9 in S1 Text.

**Deaths in the community.**   The data on community-based deaths—in other words, the deaths outside of health facilities' purview—is too sparse to develop splines as above. As an exercise to understand the total magnitude of this burden, albeit not the age-specific magnitude, we have followed the procedure outlined by Shi and colleagues [30]. The number of inpatient deaths was multiplied by 1.5 to 2.9, which were estimates from 3 datasets of the total ALRI mortality as a factor to healthcare attended ALRI mortality (see Shi and colleagues [30] for the data). The mortality was then further multiplied by a factor of 0.9 to 1.0, which accounts for the possible proportion of influenza-related ALRI that may be misclassified as a result of co-occurring influenza and RSV seasons in the 3 studies (see Shi's supplementary tables 22 to 24 in pages 65 to 66) [30]. The more recent method of Li and colleagues also could not do age profiling from the data that looked at the proportion of all-cause deaths that were attributable

to RSV, as these data were only available from the group of 0 to 60 month olds as a single age group.

**Scenario analysis: Severity of RSV.** There is an incomplete picture of how hospitalization and subsequent death are related to the incidence of severe RSV disease—defined by the WHO's Integrated Management of Childhood Illnesses as the presence of chest indrawing, difficulty breathing, or very severe in the presence of canonical pediatric danger signs of distress [38,39] (further explanation on what constitutes a severe or very severe infection is found in Section S1.8 in S1 Text). We found no study in the literature that could describe the impact of severity on the entire cascade of care as depicted in Fig 1; in particular, we did not find the probability of death conditional on severe or very severe categorization within the community or the hospital. Our model of severity from the community- or hospital-based incidence is shown in Fig B in S1 Text. There are indications that some hospitalizations are non-severe, and some severe cases do not access care [30].

Therefore, for our main analysis, we chose hospitalization as a dual metric of the severity of disease as we are interested in the resource use expended on RSV-attributable disease. However, we applied our approach to the available data on the probability of severe and very severe cases in the community and in the hospital as this could be constructive, and these results are presented in the appendix.

## Results

To estimate the relationship between age and RSV cases, hospitalization, and mortality, we used 18 studies of community-based incidence, 37 studies of hospital-based incidence, 8 studies for the probability of hospitalization among cases in the community, and 58 studies for hCFR among hospitalized patients (Table 1). These studies presented 38,488 person-months of observation 1,943 cases in the community, almost 5 M person-months of observation and 27,822 cases in the hospitals, 121 hospital admissions among 609 cases in the community, and 347 deaths among 27,467 inpatient cases; further details and the number of age groups are in Table A in S1 Text.

We analyzed the incidence of "severe" and "very severe" cases separately based on 15 and 4 community-based studies and 15 and 13 hospital-based studies, respectively (Table 1). Further details on person-years of observation, events, and age groups are in Table A in S1 Text.

When we stratified these studies by World Bank income group, we found that there was a dearth of information for LICs, including no studies to examine the burden from community-based studies nor the probability of hospitalization. Among studies with data on 3 or more age groups, there were 2 studies in LICs reporting hospital-based incidence (reporting both cases and catchment population) and 9 studies in LICs reporting hospital-based deaths. In LMICs and UMICs, there was more data for all categories.

### Splines of incidence and probabilities

We tested for the statistical significance of World Bank income group as a predictor of the outcome (Table 2). The income group was a significant predictor for incidence only for community-based incidence (Splines I). Income group was not significant for hospital-based incidence (Spline II), the probability of hospitalization (Spline III), or of death among hospitalized cases (Spline IV).

The splines for each outcome are shown in Fig 2. We have opted for graphical representations of the splines rather than tables because GAMs do not have an explicit, closed-form; the R objects for each spline are provided in **S1 Data** along with code for interested users.

**Table 2. Model selection via the generalized likelihood ratio test.**

| | Income group FE | Income group FE and RE |
|---|---|---|
| Spline I: Community-based incidence | P = <0.01 (DF = 2, $\chi^2$ = 13.67) | P = 0.6 (DF = 3, $\chi^2$ = 1.85) |
| Spline II: Hospital-based incidence | P = 1 (DF = 4, $\chi^2$ = 0) | P = 0.16 (DF = 7, $\chi^2$ = 10.53) |
| Spline III: Probability of hospitalization | P = 0.86 (DF = 2, $\chi^2$ = 0.3) | P = 0.82 (DF = 3, $\chi^2$ = 0.91) |
| Spline IV: Probability of death among hospitalized cases | P = 0.63 (DF = 4, $\chi^2$ = 2.56) | P = 0.14 (DF = 7, $\chi^2$ = 10.9) |

DF, degrees of freedom; FE, fixed-effects; RE, random-effects.

**Incidence in community- and hospital-based studies.** We found that the incidence among community-based studies is lowest among newborns, peaks before 6 months of age, and then decreases, and the overall incidence is higher in UMICs than in LMICs (Fig 2). As there were no studies among LICs and only 6 studies among UMICs in the training dataset, the global trend (in pink) is driven by the trend among LMICs, for which there were 12 studies in the training dataset. Among neonates (<28 days of age), the incidence is 14 (95% CI: 3, 35) per 1,000 person-years in LICs and LMICs, and 68 (95% CI: 13, 217) person-years in UMICs. The peak incidence is 108 (95% CI: 67, 166) per 1,000 person-years at 5.3 (95% CI: 3.5, 7.3) months in LICs and LMICs, and 106 (95% CI: 53, 187) at 5.8 (95% CI: 0.1, 60.0) months per 1,000 person-years in UMICs (Fig 2). The peak incidence at UMICs was difficult to ascertain because of the heterogeneity of age profiles across countries.

Among hospital-based incidence studies, we find a similar pattern but with an earlier peak than community-based incidence, and the incidence is lowest after the age of 2. The neonatal and peak hospital-based incidence is about a tenth as high as that of the community-based incidence. Among neonates (<28 days of age), the incidence is 5 (95% CI: 3, 11) per 1,000 person-years in all regions. The peak of hospital-based incidence is 24 (95% CI: 15, 36) per 1,000 person-years in all regions.

**Probability of hospitalization among cases in the community.** The highest probability of hospitalization is among newborns, at 25% (95% CI: 13%, 41%), decreasing to 13% (95% CI: 7%, 20%) by 6 months of age, 11% (95% CI: 6%, 17%) by 12 months of age, and 7% (95% CI: 3%, 13%) by 5 years of age (Fig 2).

**Case fatality rates among hospitalized individuals (hCFR).** The highest probability of hospital-based case fatality rate (CFR) is among neonates, at 1.27% (95% CI: 0.65%, 2.18%), decreasing to 0.64% (95% CI: 0.40%, 0.95%) by 6 months of age, 0.53% (95% CI: 0.33%, 0.81%) by 12 months of age, and 0.64% (95% CI: 0.19%, 0.62%) by 5 years of age (Fig 2).

**Within-sample and out-of-sample validation.** Randomly sampled curves drawn from the spline models were graphed against the data that was used to estimate the models (see S2-1.1 to S2-1.4 in S2 Text) and against the studies used for out-of-sample validation (see Sections S2-2.1 to S2-2.4 in S2 Text). Visual inspection revealed no overwhelming lack-of-fit nor signaled that our specifications were inappropriate.

## Age profile of RSV burden

We have combined the splines (Fig 2) as described in Fig 1 and in Section S1.4 in S1 Text to characterize the number and the age profile of RSV cases per 1,000 person-years in the community, in hospitals, and as in-hospital deaths (see Fig 3). To describe the age profile of RSV burden (Fig 3), we show the peak, median, and mean age of each outcome in Fig 4. Overall,

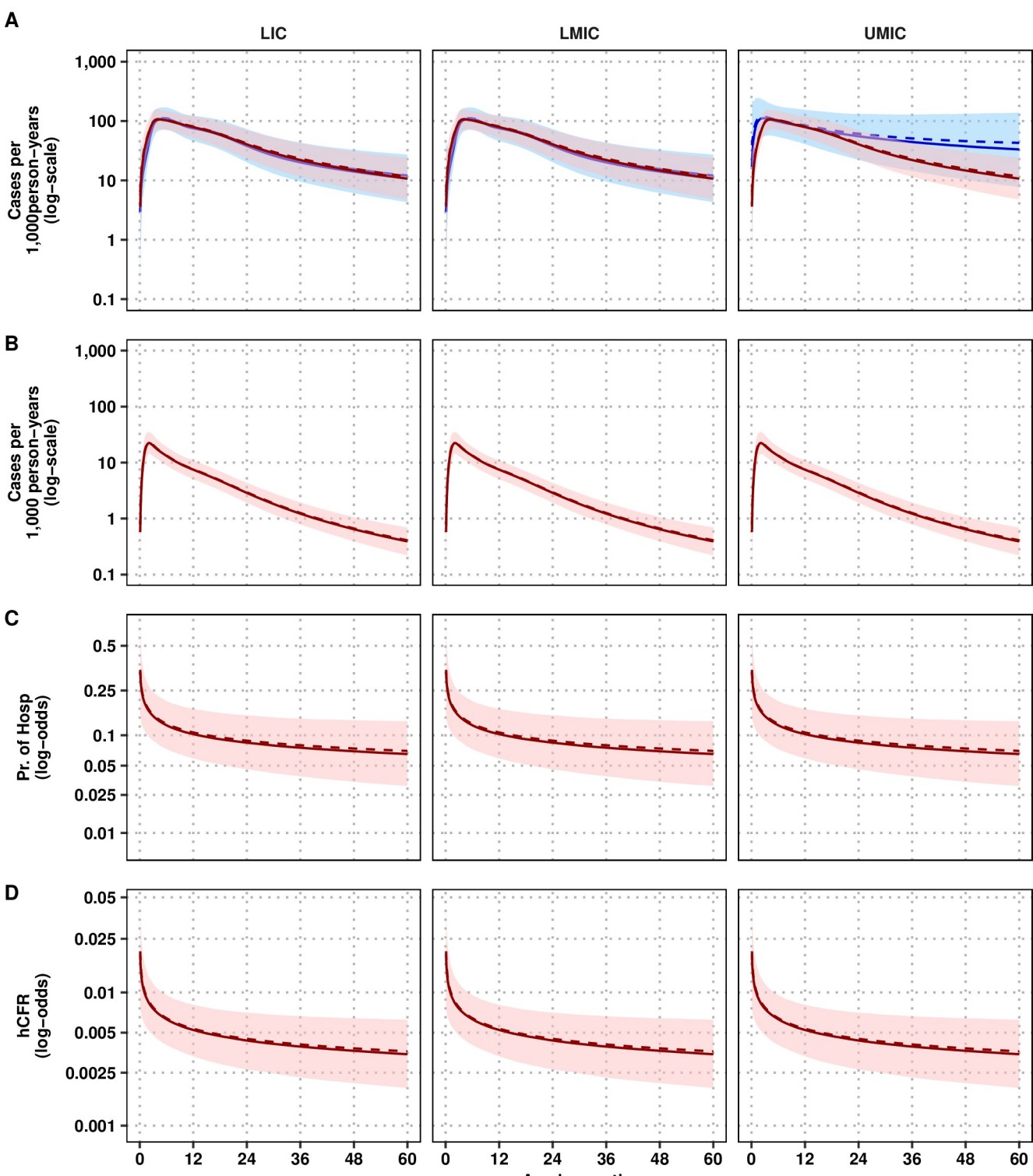

**Fig 2.** Splines of (A) community-based incidence, (B) hospital-based incidence, (C) probability of hospitalization, and (D) probability of death among hospitalized cases by income group designation. The bands correspond to the 95% confidence intervals of each parameter at each age. The pink bands present the "global" splines—derived from GAMMs with no income group predictor—and the blue bands present the "income group" splines—derived from GAMMs that include a predictor for the World Bank's country income group designation. For the community-based incidence spline and the probability of hospitalization spline, the LIC estimates are identical to the LMIC estimates because no data exists from LIC settings. GAMM, generalized additive mixed model; hCFR, hospital-based case fatality rate; LIC, low-income countries; LMIC, lower-middle-income countries; UMIC, upper middle-income countries.

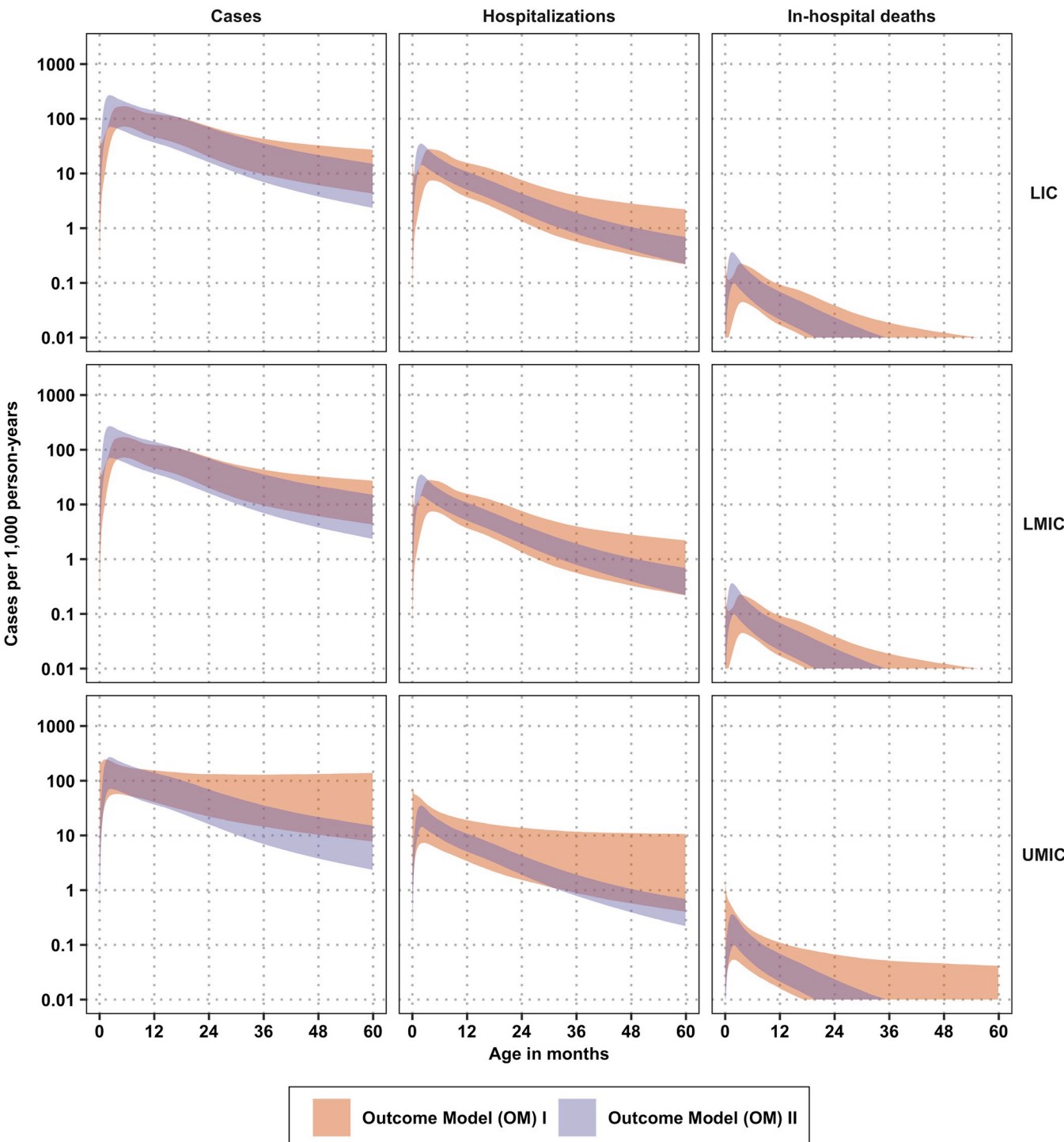

**Fig 3. RSV cases, hospitalizations, and deaths per 1,000 person-years according to OM I and II.** 95% confidence intervals of cases, hospitalizations, and deaths per 1,000 person-years according to OM I and II, detailed in Fig 1 and Section S1.4 in S1 Text. The orange bands arise from OM I, calculated by taking the splines of community-based incidence (Spline I), probability of hospitalization (Spline III), and probability of death among hospitalized cases (Spline IV). The purple bands arise from OM II, calculated by taking the splines of hospital-based incidence (Spline II), probability of hospitalization (Spline III) to back-calculate cases in the community, and probability of death among hospitalized cases (Spline IV). LIC, low-income countries; LMIC, lower-middle-income countries; OM, outcomes model; RSV, respiratory syncytial virus; UMIC, upper middle-income countries.

the incidence of any outcome had more uncertainty when the incidence came from community-based data (Spline I; OM I) than when the incidence came from hospital-based data (Spline II; OM II).

In all regions, the metrics of the age profile (the mean, median, and peak age) of cases in the community were later than that of hospitalizations and deaths; therefore, both models confirm the hypothesis that more severe outcomes have a younger age profile than RSV cases in general (Fig 4). Throughout the regions, the peak of all outcomes was earlier than the median age of the disease, which itself was lower than the mean age of the disease. All metrics of the age profile of all outcomes overlapped markedly between OM I and OM II, indicating that the uncertainty within models was larger than the uncertainty between models. However, the distribution of the window tends to be lower in OM II than in OM I, indicating that hospital-based incidence studies will bias the age profile of RSV outcomes down.

To quantify the proportion of cases that are contained within key age groups targetted by prophylactics in development, we have quantified the proportion of cases under the age of 5 that fall in age-groups of 3 months in the first year of life in Fig 5. The model based on community-based incidence indicates that a lower proportion of hospitalizations and deaths occur in the 0–<3 month olds, who bear 11% of hospitalizations and 18% of deaths under 5 years of age, than in the 3–<6 month olds, who bare 17% of hospitalizations and 20% of deaths under 5. However, this relationship is reversed in the model based on hospital-based incidence, where 0–<3 month olds account for 20% to 29% of hospitalizations and deaths under 5 and 3–<6 month olds account for 19% to 21% of all hospitalizations and deaths under 5. The difference between models is not statistically significant (Fig A in S3 Text), and there is no evidence that the burden in infants 0–<3 months and 3–<6 months of age is different. Together, these figures indicate that children under 6 months of age—who make up 10% of the population of children under 5 (see Table B in S1 Text)—bear 20% to 29% of cases, 28% to 39% of hospitalizations, and 38% to 50% of deaths.

**Severe and very severe cases.** The incidence of severe and very severe cases in community-based and hospital-based surveillance was modeled against age (Figs B and C in S3 Text). While the country-level income group was not a significant predictor for the probability of severe and very severe cases in the community or severe disease in hospitals, the country-level income group was a significant predictor of the probability of very severe disease in hospitalized cases (see Table A in S3 Text). For the probability that hospitalized cases are very severe, the model shows a similar mean for LIC and LMIC, but more uncertainty among LMICs, and in UMICs, the model shows a lower distribution and more uncertainty that hospitalized cases are very severe (see Fig B in S3 Text). The incidence of very severe cases among hospitalizations is different across country income groups due to both the underlying incidence in cases and the probability that those cases turn out to be very severe (see Fig C in S3 Text).

There were no statistically significant differences between the mean, median, and peak age of severe and non-severe disease within each model, although the point estimate and the uncertainty bounds indicate that severe cases have a younger age distribution than non-severe cases (compare Fig D in S3 Text to Fig 4). While children under 6 months of age make up 10% of the population of children under 5 (Table B in S1 Text), the share of severe cases borne by this age group is 30% to 42% and 31% to 42% of all severe and very severe cases in the community, and 32% to 44% and 33% to 45% of the severe and very severe cases in hospitals. While the severity as a proportion of community and hospital cases was modeled, the ultimate outcomes of severe cases (convalescence or death) could not be modeled because these relationships were not clearly reported in the literature [30,31].

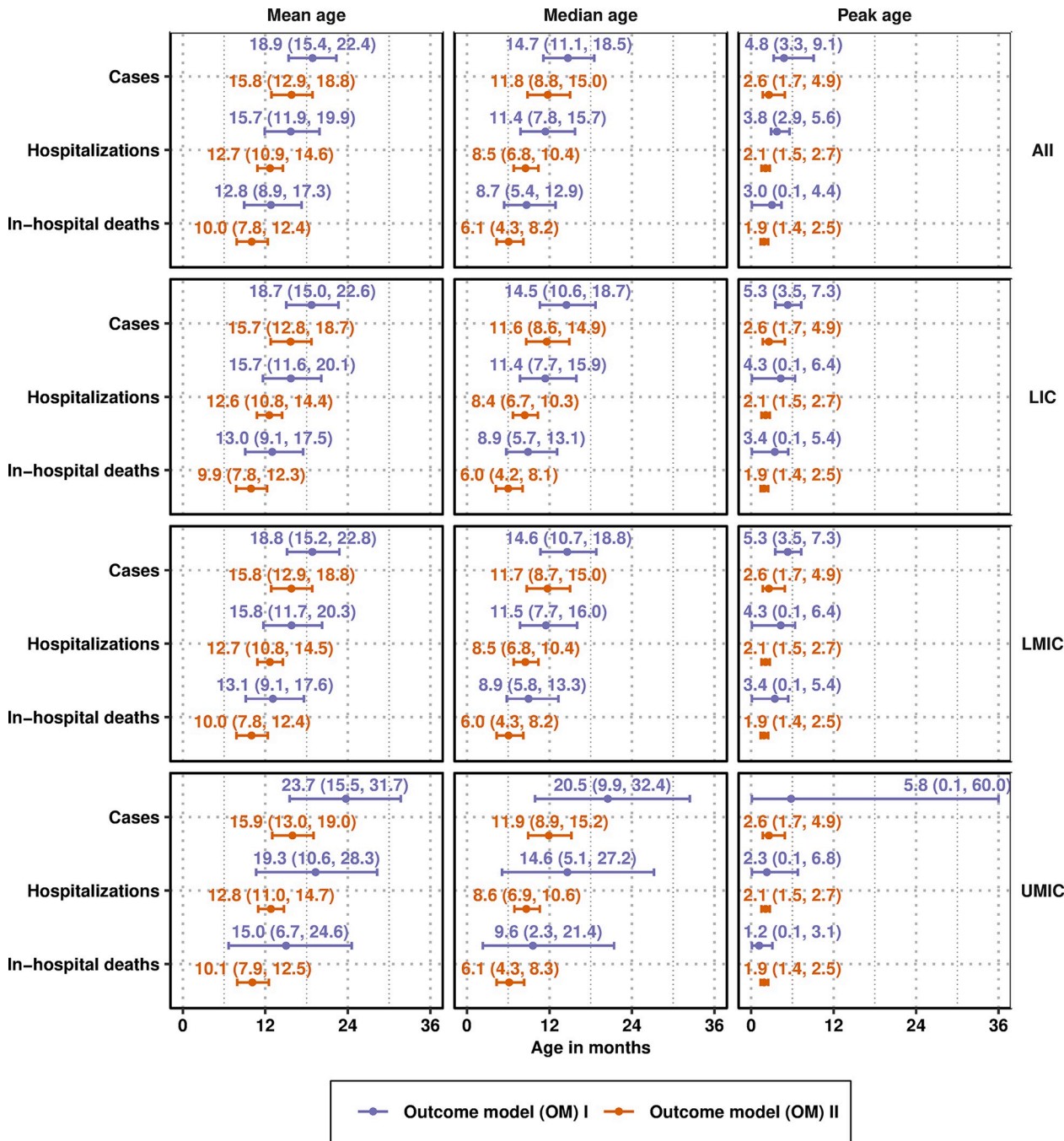

**Fig 4. Mean age, median age, and peak age of infection, hospitalization, and deaths due to RSV.** The segments correspond to 95% confidence intervals of each of the summaries and the dots correspond to the median estimate of each of the summaries; these are shown in orange when calculated using OM I and in purple when calculated using OM II. LIC, low-income countries; LMIC, lower-middle-income countries; OM, outcomes model; RSV, respiratory syncytial virus; UMIC, upper-middle-income countries.

## Burden estimates

Among the 27 LICs, 53 LMICs, and 41 UMICs in our analysis, the population of children under 5 years of age numbered 103 million, 335 M, and 161 M in LICs, LMICs, and UMICs, respectively, for a total of 598 M children under 5. Collectively, there were 61 M children

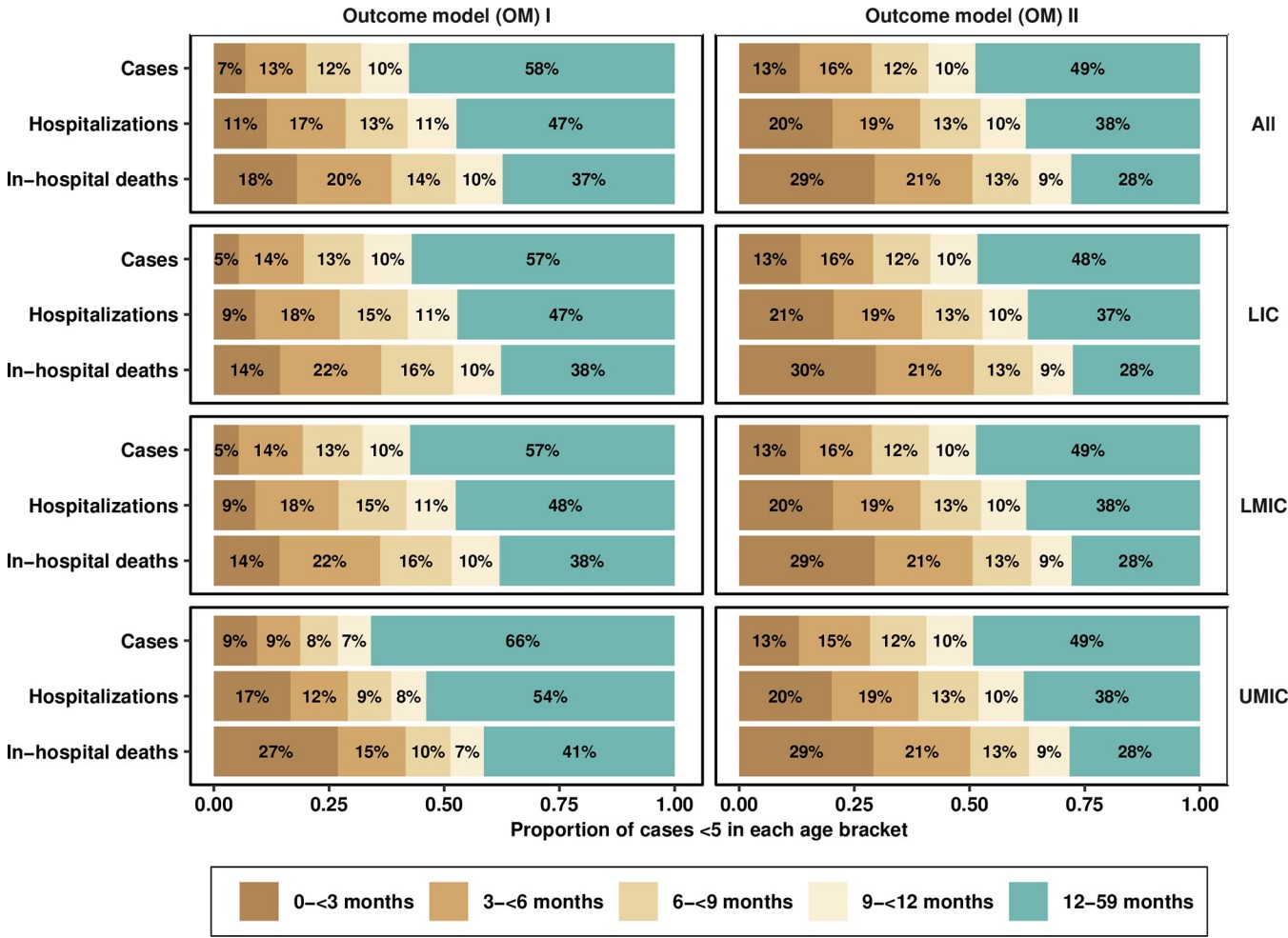

**Fig 5. Proportions of each outcome that fall under key age brackets.** Uncertainty intervals are presented in Fig A in S3 Text. LIC, low-income countries; LMIC, lower-middle-income countries; OM, outcomes model; UMIC, upper middle-income countries.

under 6 months of age, 60 M children between 6–<12 months of age, and 468 M children between ages 1–<5 years (a cross-tabulation of the population by income group and age is found in Table B in S1 Text). Fig 6 shows the burden of disease in each income region, and Fig 7 shows the burden of disease by age group for each OM and BM combination.

The point estimates of the total number of cases in all low- and middle-income countries across all models (BM I/OM I, BM I/OM II, BM II/OM I, and BM II/OM II) ranged between 24.83 M (95% CI: 12.20 M, 44.43 M) and 31.34 M (95% CI: 28.92 M, 34.05 M) cases (Fig 6). Estimates for hospitalizations ranged between 2.62 M (95% CI: 1.81 M, 3.68 M) and 3.57 M (95% CI: 1.98 M, 5.83 M), and estimates for in-hospital deaths ranged between 16,326 (95% CI: 9,018, 26,972) to 22,229 (10,328, 41,763). Furthermore, the uncertainty within each model surpasses the uncertainty across models. Notably, there is more variance in each outcome with BM I/OM II than with BM I/OM I, attributable to the greater variance in the hospital-based incidence studies, which are the basis for OM II (see Figs 2, 3 and 6).

**Burden stratified by income group.** According to BM I, the plurality of cases are in LMICs, followed by UMICs, followed by LICs. We used the country-specific burden model estimates developed by Shi and colleagues and updated by Li and colleagues (the basis for BM

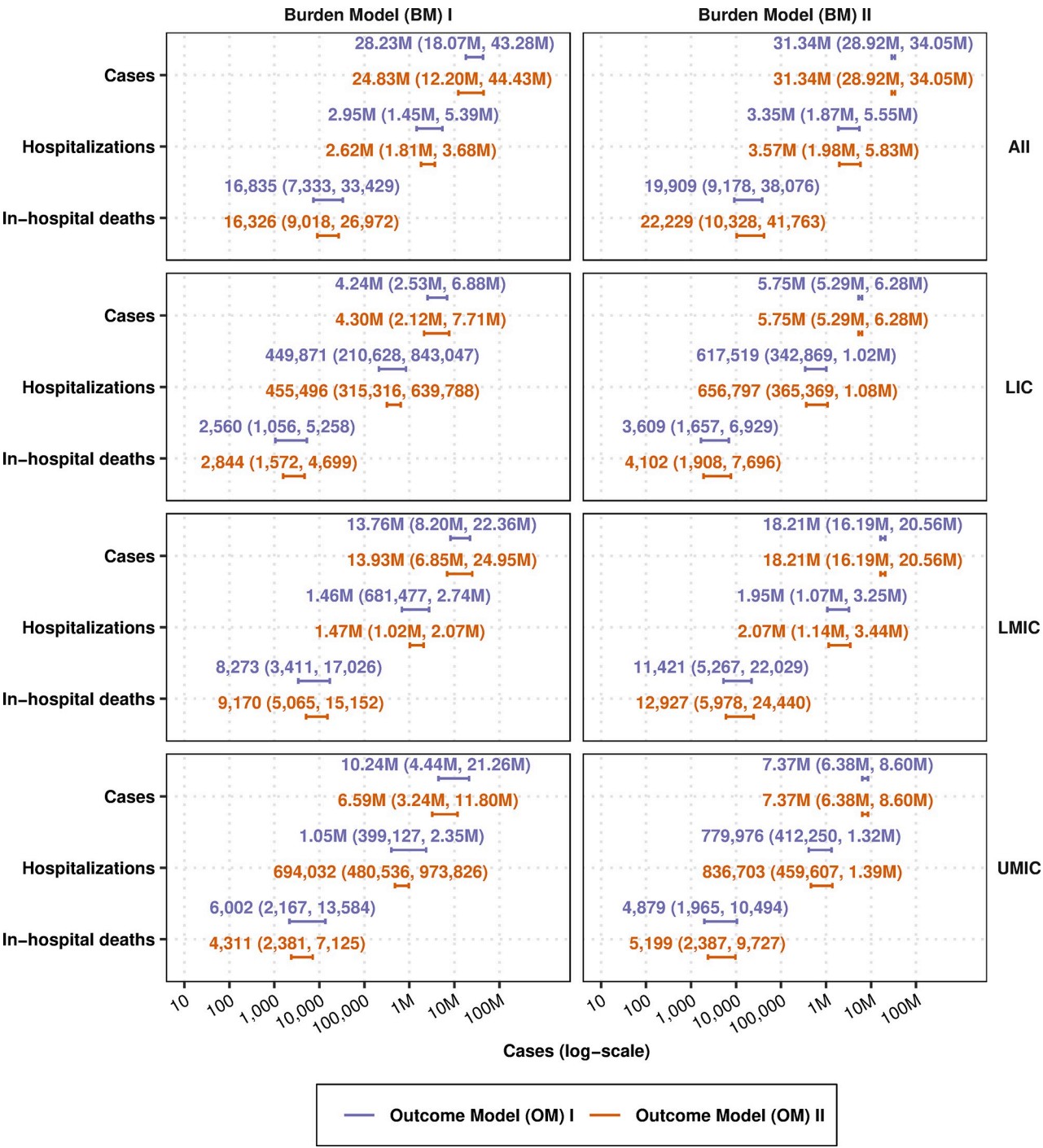

**Fig 6. Burden of RSV cases, hospitalizations, and deaths in low- and middle-income countries, by World Bank income group classification.**
Number and 95% confidence intervals of cases, hospitalizations, and deaths per 1,000 person-years according to OMs I and II and BMs I and II, as detailed in Fig 1 and Sections S1.4 and S1.6 in S1 Text. Because we used Li's country-specific model as the basis for the number of cases in BM II, the cases do not differ between OM I and OM II; hospitalization and in-hospital death outcomes differ due to the different ways in which conditional probability splines were applied in OM I vs. OM II. Burden stratified by age group can also be found in the supplement (see S3 Text, Figs F–H). BM, burden model; LIC, low-income countries; LMIC, lower-middle-income countries; OM, outcomes model; RSV, respiratory syncytial virus; UMIC, upper-middle-income countries.

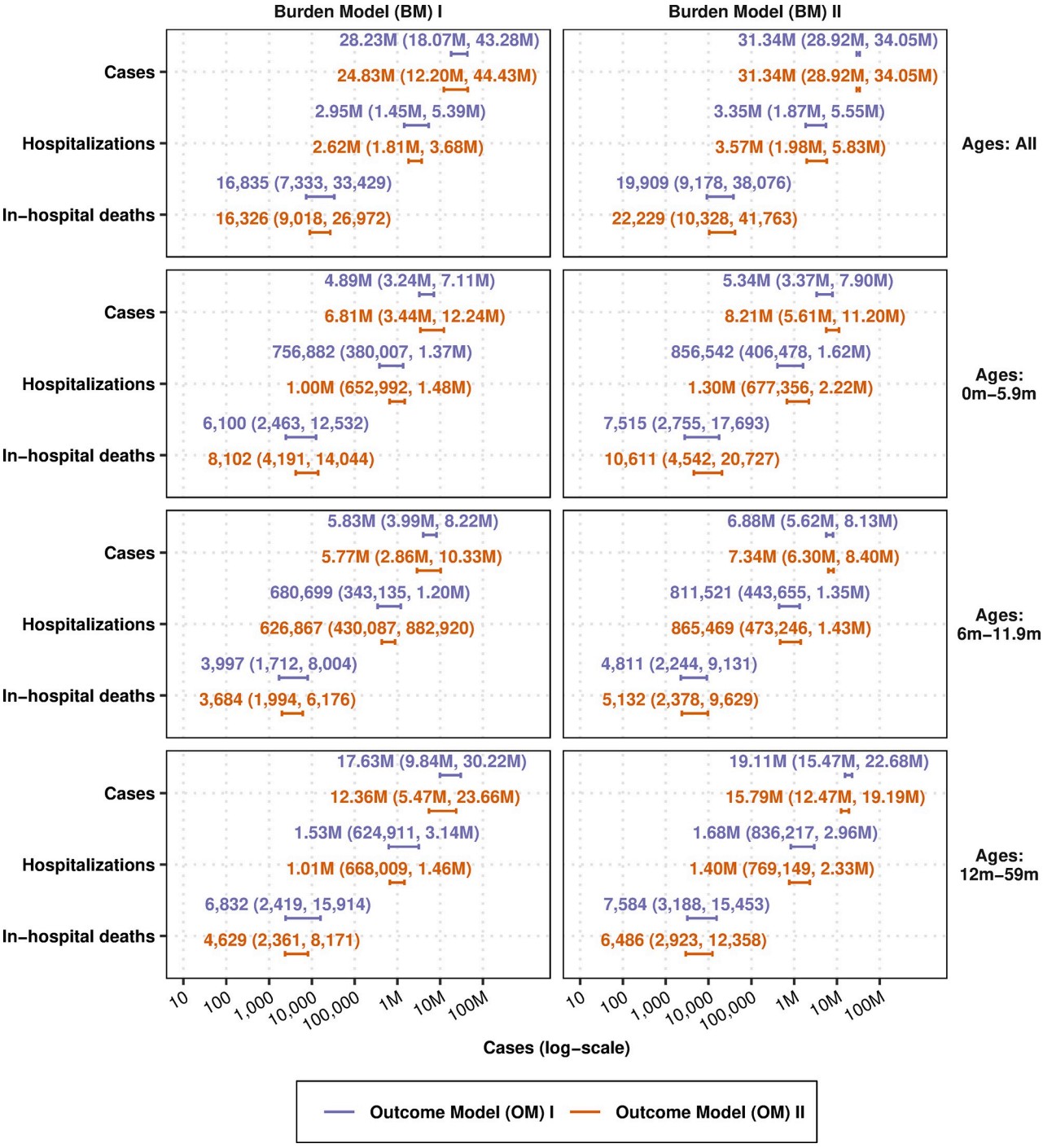

**Fig 7. Burden of RSV cases, hospitalizations, and deaths in low- and middle-income countries, by age group.** Number and 95% confidence intervals of cases, hospitalizations, and deaths per 1,000 person-years according to SMs I and II and BMs I and II, as detailed in Fig 1 as well as Sections S1.4 and S1.6 in S1 Text. Burden stratified by age group can also be found in the supplement (see Figs F–H in S3 Text). BM, burden model; RSV, respiratory syncytial virus; SM, spline model.

II/OM I and BM II/OM II). The trends in BM II are similar to those of BM I: the plurality of cases are in LMICs, followed by UMICs, and lastly by LICs. Because their model did not estimate other outcomes (hospitalization or hCFR), the estimates of these outcomes differ

between BM II/OM I and BM II/OM II although both outcome models begin by assuming same number of total cases (see Fig 6, right column).

**Burden stratified by age group.** The different models predict cases ranging from 24.83 M to 31.34 M, and out of those, from 4.89 M to 8.21 M cases will occur in the first 6 months of life, while another 5.77 M to 7.34 M cases will occur in the ages of 6 to 11.9 months (see Fig 7). The country-specific burden model by Shi and colleagues (and more recently by Li and colleagues) did not assign cases to any age group; therefore, the distribution of cases among age groups differs between BM II/OM I and BM II/OM II even if the total number of cases does not. Our models then yield between 752,882 to 1.30 M hospitalizations in the first 6 months of life and between 626,867 and 865,469 hospitalizations between the ages of 6 to 11.9 months. In-hospital deaths were then estimated at 6,100 to 10,611 in the first 6 months of life and between 3,684 and 5,132 in the ages of 6 to 11.9 months.

**Burden stratified by age group and income group.** The incidence stratified by both age and country income group is shown in Figs F to H in S3 Text. All regions follow similar age-related patterns. Country-specific burden estimates by month of age for OMs I and II and BMs I and II assumptions can be found in S1 Data where one can find 1 spreadsheet for each country.

**Estimates of deaths in the community and total deaths.** Although there was no data to look at the deaths outside of healthcare by age, and therefore there was no data to construct splines, we performed a simple estimate of the number of total deaths—both inside and outside of healthcare facilities (see Table B in S3 Text). Overall, we estimate a mean of between 34,116 total deaths (BM I/OM II) and 46,485(BM II/OM II) deaths, but the uncertainty within the models outpaces the uncertainty between models, featuring confidence intervals as low as 13,589 (BM I/OM II) and as high as 94,423.

## Discussion

Understanding the burden of RSV disease across LICs, LMICs, and UMICs is possible thanks to the extensive amount of data that has been collected across the literature and the work of systematic reviews like that of Shi and colleagues and updated by Li and colleagues [30,31]. Yet, not all these data are amenable to synthesis and the data sometimes point to distinct patterns that must be interrogated in order to inform policy. In this paper, we have set forward a principled framework to leverage the extant data.

Constructing splines from outcomes of community- and hospital-based incidence, probability of hospitalization, and in-hospital death, there is enough evidence to show that community-based incidence differs by World Bank Income Group, but there is no statistically significant evidence for a difference by World Bank Income Group for hospital-based incidence, probability of hospitalization, or the probability of in-hospital death ($p \leq 0.01$, $p = 1$, $p = 0.86$, $0.63$, respectively). In supplemental analysis—examining the probability of severe and very severe disease among cases in the community and in the hospital—we found that there is enough evidence to adjust splines for World Bank Income Group when modeling very severe disease in hospitalized cases ($p \leq 0.01$) but not other outcomes (see Table A in S3 Text).

The peak age of community-based incidence is 4.8 months, and the mean and median age of infection is 18.9 and 14.7 months, respectively (Fig 2). Estimating the age profile using the incidence coming from hospital-based studies yields a slightly younger age profile, in which the peak age of infection is 2.6 months and the mean and median age of infection are 15.8 and 11.6 months, respectively. Children under 6 months of age bear 20% to 29% of cases, 28% to 39% of hospitalizations, and 38% to 50% of deaths despite constituting only 10% of the population under 5. Moreover, this share is a bit more concentrated among children 3–<6 months

old than among those 0–<3 months old for cases and hospitalizations—likely as maternal-derived immunity wanes, although deaths are more concentrated in 0–<3-month-olds rather than the 3-<6 month-olds (18% to 29% versus 20% to 21%, respectively, Fig 5).

More severe outcomes, such as hospitalization and in-hospital death have a younger age profile. Unlike the results of the model without severity (Fig 4), the age profile of severity does not seem to be younger in the hospital than in the community when restricting to the population of severe and very severe disease (Fig E in S3 Text). Children under 6 months of age bear about the same proportion of severe cases as hospitalized cases overall; they bear 30% to 42% and 31% to 42% of all severe and very severe cases in the community, while they bear 32% to 44% and 33% to 45% of the severe and very severe cases in hospitals.

Throughout all low- and middle-income countries, there were between 21.83 M to 31.34 M cases, between 2.62 M and 3.57 M hospitalizations, and between 16,326 to 22,229 in-hospital deaths. Overall deaths—in the community as well as in the hospital—were estimated at 34,111 to 46,485, although the uncertainty within models is larger than the uncertainty between models. The largest share of cases were in LMICs, followed by UMICs, and lastly by LICs.

Modeling semi-parametric splines provides more accurate estimates of per-month incidence than conventional meta-analyses of the age-specific incidence reported in the literature. Semi-parametric splines leverage literature from the field where different age group breakdowns are presented. The combination of a semi-parametric approach and simulation was able to quantify the mean, median, and peak age of all outcomes along with their uncertainty in order to stimulate conversations to define the key window of protection. While the focus on the overall burden is important, an additional target metric ought to be the capacity of the prophylaxis to protect children through the key window of infection, hospitalization, and death. One advantage of our approach is the capacity to present the percentage of children that could be protected by products of different durations (Fig 5).

To understand the impact of different ways of interpreting the data and the extant estimates of burden (from Shi and colleagues' and Li and colleagues' risk factor model), we devised 2 outcome models (OM I and OM II) and applied each of them to 2 burden models (BM I and BM II), for a total of 4 models listing cases in the community, hospitalizations, and in-hospital deaths. Because there is no way to weigh these models in a statistically principled manner (unless one arbitrarily assigns equal weights), we did not combine the models. Though prediction intervals between the models overlap substantially (Figs 6 and 7), indicating that both methods of assessing burden were not significantly different at a level of 95% confidence, distributions with a lower or higher mean might be enough to influence a cost-effectiveness analysis. Our approach also examined the robustness of the assumption that a country's income group is an adequate proxy to interpolate epidemiological patterns of RSV for countries that lack data. We found that the country income group was only useful for community-based incidence (Table 2).

Our findings imply that prophylactics that offer protection until 4 to 6 months of age will target the peak age of deaths but are unlikely to avert even 50% of deaths (indicated by the median in Fig 4 and Fig 5). The longest lasting form of passive immunity—in the form of a monoclonal antibody, Nivursimab—so far protects for 150 days or 5 months [19]. As protection from such a product wanes in the fifth and sixth month, our findings indicate that a second product would be helpful as well. This is consistent with cost-effectiveness analyses, which show that the use of a single prophylactic must be priced competitively in order to prove cost-effective in most low- and middle-income settings [21,24]. While the data on severity was less clear and was relegated to supplemental analysis, severe RSV disease showed a higher concentration in younger infants as well (Fig D and E in S3 Text).

Notably, Li and colleagues found that the RSV-associated acute lower respiratory infection incidence rate peaked in children aged 0–<3 months in LICs and LMICs, whereas the rate peaked in children aged 3–<6 months in UMICs. This tends to be younger than our point estimates based on community-based incidence (OM I) but not hospital-based incidence (OM II), presumably because hospitalized cases are younger than the average case observed in the community. However, it should be noted that Li an colleagues calculated the incidence in 3-month age bands for the first year, which may lead to a masking of the increases and decreases in those first months of life, a shortcoming that our approach has overcome. For cost-effectiveness analysis for which conclusions are reached on the balance of all uncertainties in parameters, the lower age profile could change decisions, especially due to shifts in mortality, which accounts for the majority of all disability-adjusted life-years. In particular, Shi and colleagues found that 45% of deaths appear to occur in the first 6 months of life, which agrees with the estimates of Li and colleagues (45,000 deaths among 0–<6 months compared to 100,500 among 0 to 60 months), and also agrees with our estimates of 38% to 50% (Fig 7). We showed, however, that accounting for model assumptions casts doubt on estimating a mean age of death at 6 months of age, which depends on trusting data coming from hospital-based incidence studies over the data coming from community-based studies (Figs 4 and 5).

Our results are similar to those of Nyiro and colleagues [35], who used a catalytic model of infection on data on RSV-binding IgG serostatus. Nyiro and colleagues showed that the mean age of infection is 15 months (95% CI 13, 18 months) and that the most vulnerable period for infection was between the time when maternal antibodies wane, at 4.7 months, and 11 months of age. Like Nyiro and colleagues [35], we found that the peak age at infection ranged between 3.3 months and 5.8 months (Fig 4) across income groups.

Another study focused on determining the mean age of death from RSV by online questionnaire from Scheltema and colleagues [34] estimated median ages of 5.0 (IQR: 2.3 to 11.0) months in LICs, 4.0 (IQR: 2.0, 10.0) months in LMICs, and UMICs: 7.0 (95% CI: 3.6, 16.8) months. While our estimates do not differ in a statistically significant manner against those of Scheltema and colleagues, the distribution of our estimate of median age of death is slightly higher: 8.9 (95% CI: 5.7, 13.1) and 6.0 (95% CI: 4.3, 8.1) in LICs, 8.9 (95% CI: 5.8, 13.3) and 6.0 (95% CI: 4.3, 8.2) in LMICs, and 9.6 (95% CI: 2.3, 21.4) and 6.1 (95% CI: 4.3, 8.3) in UMICs (Fig 4).

Overall, our conclusions and those of Li and colleagues' [31] came to qualitatively similar conclusions regarding the overall burden of disease. Li and colleagues' [31] generalized linear mixed model (GLMM)—similar to our GAMMs but without the versatile splines—calculated 31.4 M (23.2 M, 42.6 M) cases of RSV in countries designated as "developing" by UNICEF (closely equivalent to the World Bank's LICs, LMICs, and UMICs together), which was close to our estimates between 24.67 M to 28.23 M in BM I (which, unlike BM II, was not based on their risk factor model). Among hospitalizations, our estimates differed more markedly but were in the same order of magnitude; while they estimated 3.1 M (95% CI: 2.4 M, 4.2 M) hospitalizations in developing countries, we estimated between 2.95 M and 3.57 M cases across models, with uncertainty ranges between 1.45 M and 5.83 M. Our in-hospital deaths arrived at comparable conclusions to those of Li an colleagues: our point estimates ranged from 16,326 to 22,229 compared to their 25,900 and showed similar credible intervals—7,333 to 41,763 compared to their 14,500 to 48,600 (Table B in S3 Text) [31].

One key difference between the burden models is that our total deaths—both in-hospital and in the community—34,116 to 46,485 lie much closer to those of GBD 2015 estimate of 36,400 (95% CI: 20,000, 61,500) for developing countries than those of Li and colleagues at 100,500 (95% CI: 83,600, 124,500) and Shi and colleagues of 117,964 (95% CI: 94,545, 147,164) [30,31,40]. Shi and colleagues estimated a higher hCFR in neonates (≤28 day olds) of 6.3%

(3.3%, 12.1%) and 1.0% (95% CI: 0.1%, 7.2%) (see their supplementary table 20) than ours, which were 1.28% (95% CI: 0.66%, 2.22%) (Fig 2). For the age group 0–<6 months, Shi and colleagues calculated 2.2% (95% CI: 1.8% to 2.7%) for all developing countries (loosely equivalent to LICs, LMICs, and UMICs), whereas Li and colleagues' update and reanalysis estimated an hCFR of 1.1% (95% CI: 0.8%, 1.6%). Li and colleagues also estimated 1.9% (95% CI: 1.5%, 2.4%) in sensitivity analyses. Therefore, we suspect that part of the difference between our study and that of Shi and colleagues is that they used only studies that showed neonates (≤28 day olds) specifically and, unlike our framework, had no framework to "borrow" or inform their estimate of ≤28-day mortality with the implicit information from studies for children 0–<3 months or 0–<6 of age (see their appendix page 62). Our framework allowed for "borrowing" between incomparable age breakdowns in order to be more precise about our final estimate of hCFR [30,31].

Despite the ambitious efforts of RSV GEN led by Shi and colleagues [30], and more recently by Li and colleagues [31], key gaps remain in the literature that our analysis could not address. Because we required incidence estimates from 3 nonoverlapping age groups in order to calculate the splines, there was no calculation of community-based incidence in LICs. Therefore, the assessments of incidence in this country group must rely on hospital-incidence studies, leaving a formidable question around the cases in the community in LICs and underestimating the impact of future prophylactics in these settings. Second, data on the hospitalization probability among cases in the community are sparse in all country groups, and, therefore, it is difficult to understand how hospitalization incidence compares to community incidence; community-based mortality studies have shown a lower median age than hospital-based mortality [41]. Third, severity data was difficult to interpret, but the patterns indicate that the mean and median age of severe infection is situated at a younger age than the mean and median age of all cases, consistent with the main model (see Figs D and E in S3 Text versus Fig 4). Severity itself is not always defined the same way, and therefore, the continuum between a mild case, a severe case, and a very severe case is not entirely clear (see Section S1.8 in S1 Text). Lastly, the existing heterogeneity between World Bank income groups detected for severity in our analysis pointed to the need for more geographically comprehensive surveillance, in particular, with regard to community-based incidence, severity, care-seeking, and hospitalization in LICs.

Different demand and accessibility characteristics by income group could explain why the incidence—both from the community and hospital-based studies—was found to be higher in UMICs than in LMICs or LICs (see Fig 2). LIC hospital incidence is likely underestimated because patients do not reach health facilities, or when they do, they have to be turned away due to overcapacity [42]. Many of the original studies retrieved by Shi and colleagues, as well as Li and colleagues, were previously unpublished (now available at [43,44]), and the circumstances under which patients were admitted are unknown, thus leaving the degree of underestimation unclear. Given the positive association between country-level income and healthcare accessibility, the overall burden may have been understated in poorer locations. Healthcare use surveys alongside RSV incidence studies would help estimate the size of the underreporting, as is done for other diseases [45].

Four last gaps remain in the data we examined. First, subtype identification (RSV A versus RSV B) was so poorly characterized in the extant literature as to disallow a meta-analytic analysis like the one performed here. If prophylactics are not equally effective against both subtypes, then further surveillance on the incidence of subtypes would be necessary. Second, there was no seasonality examined in these data, though some policies in high-income countries are being designed to target very young children around the peak RSV season of the year. Third, the period we used for the data, i.e., studies published between 2000 and 2020, included the

introduction of pneumococcal vaccine in many countries after 2008, which may have modified the CFR patterns, but it was beyond the scope of this study to examine time variation in the CFR. Fourth, the RSV seasonality has been impacted by the COVID-19 pandemic, and continuous surveillance is essential to monitor the global RSV seasonal pattern across global regions [46–48].

To date, there are 13 pediatric vaccines in development and our analysis serves to highlight the use-case for these vaccines [20]. Although the incidence of hospitalization and death has waned by the time a child is 12 months of age, we show that between 38% and 47% of RSV-related hospitalizations and between 28% and 37% of deaths among preschool children occur over the age of 1 (Fig 5). A product that can be immunogenic in children at 9 months of age could prevent an additional 9% to 10% hospitalizations and deaths which occur between 9–<12 months when children are scheduled to begin taking childhood vaccinations for other commonly administered antigens like measles.

In conclusion, the fact that the median and mean age of infection—and of severe disease—is higher than the peak of infection indicates that an ideal strategy may be a combination of passive immunity for the first months of life followed by active immunization at 9 to 12 months of age. Although the RSV pediatric vaccine is still in the early stage of clinical development, the new long-acting monoclonal antibody (Nirsevimab, by AstraZeneca) and the RSV prefusion maternal vaccine (RSVpreF, by Pfizer) are likely to be marketed soon in high-income countries [49]. Depending on the price, the country's RSV pattern, and the implementation strategies, Nirsevimab and RSVpreF might also be able to substantially avoid RSV infections among young children in LMICs. The larger uncertainty bounds in our analysis are not a drawback of the analysis, but rather an advantage of the framework, which is steadfast in showing the uncertainty that we are confronted with in the data. As long as gaps in the epidemiological literature remain and are not examined rigorously, decision-making must be done after considering all uncertainty.

## Supporting information

**S1 Text. Supplementary Methods. Section S1.1.** Extraction of the Shi and colleagues and Li and colleagues data. **Section S1.1.1.** Modifications to incidence data from Shi and colleagues. **Section S.1.1.2.** Modifications to fatality (hCFR) data from Shi and colleagues. **Section S1.2.** Descriptive summaries of all data. **Section S1.3.** Generalized additive mixed models (GAMM). **Section S1.4.** Outcome Models (OM) I and II. **Section S1.7.** Characterizing the age profile of RSV burden (mean, median, and the peak age of each outcome). **Section S1.6.** Burden Models (BM) I and II. **Section S1.9.** Countries included, World Bank Income Group, population, and life tables. **Section S1.3.** Generalized additive mixed models (GAMM). **Section S1.5.1.** Spline specifications and computational considerations. **Section S1.5.2.** Predictions of the outcome model and uncertainty. **Section S1.8.** Severe and very severe disease. **Table A.** Characteristics of the studies included in the main splines and in the supplemental severity splines. **Table B.** Country income groups and population age <5 in 2020. **Fig A.** Geographic distribution of the available data. **Fig B.** Relationship between cases, hospitalizations, and severity from birth to 60 months of age.
(PDF)

**S2 Text. Supplementary Results: Spline Validation. Section 2–1.1.** Fit-vs-observed: Community-based incidence. **Section 2–1.2.** Fit-vs-observed: Hospital-based incidence. **Section 2–1.3.** Fit-vs-observed: Probability of hospitalization among cases in the community. **Section 2–1.4.** Fit-vs-observed: Probability of death among hospitalized cases. **Section 2–2.1.** Out-of-sample validation: Community-based incidence. **Section 2–2.2.** Out-of-sample validation:

Hospital-based incidence. **Section 2–2.3.** Out-of-sample validation: Probability of hospitalization among cases in the community. **Section 2–2.4.** Out-of-sample validation: Probability of death among hospitalized cases.
(PDF)

**S3 Text. Supplementary Results: Additional Projections. Fig A.** Splines of the probability of severe and very severe disease among community-based and hospital-based cases. **Fig B.** Splines of the probability of severe and very severe disease among community-based and hospital-based cases. **Fig C.** RSV cases, hospitalizations of severe and very severe disease per 1,000 person-years according to Spline Models (SM) I and II. **Table A.** Model selection via the generalized likelihood ratio test for severe and very severe disease. **Fig D.** Mean, median, and peak age of each severe and very severe disease among community-based and hospital-based cases. **Fig E.** Proportions of each outcome that fall under key age brackets for severe and very severe case burden. **Fig F.** Sensitivity analysis of the burden of RSV cases, hospitalizations, and deaths by age in low-income countries (LICs). **Fig G.** Sensitivity analysis of the burden of RSV cases, hospitalizations, and deaths by age in low-income countries (LMICs). **Fig H.** Sensitivity analysis of the burden of RSV cases, hospitalizations, and deaths by age in low-income countries (UMICs).
(PDF)

**S1 Data. Excel files with projections for each country of the burden of cases, hospitalizations, in-hospital and community deaths for each age group and by 1-month age groups, as well as peak, median, and mean age of each outcome.**
(ZIP)

## Acknowledgments

The authors would like to thank Philipp Tellenbach for help with data cleaning. Rescue Investigators include: Philippe Beutels (University of Antwerp); Veena Kumar (Novavax), Louis Bont (University Medical Center Utrecht); Harish Nair and Harry Campbell (University of Edinburgh); Andrew Pollard (University of Oxford); Peter Openshaw (Imperial College London); Federico Martinon-Torres (Servicio Galego de Saude), Terho Heikkinen (University of Turku and Turku University Hospital); Adam Meijer (National Institute for Public Health and the Environment); Thea K Fischer (Statens Serum Institut); Maarten van den Berge (University of Groningen); Carlo Giaquinto (PENTA Foundation); Michael Abram (AstraZeneca); Kena Swanson (Pfizer); Bishoy Rizkalla (GlaxoSmithKline); Charlotte Vernhes and Scott Gallichan (Sanofi Pasteur); Jeroen Aerssens (Janssen); and Eva Molero (Team-It Research). This publication was also supported through PATH by Gavi, the Vaccine Alliance.

## Author Contributions

**Conceptualization:** Marina Antillón, Xiao Li, Lander Willem, Joke Bilcke, Mark Jit, Philippe Beutels.

**Data curation:** Marina Antillón, Xiao Li.

**Formal analysis:** Marina Antillón, Xiao Li, Lander Willem.

**Funding acquisition:** Mark Jit, Philippe Beutels.

**Investigation:** Marina Antillón, Xiao Li, Lander Willem, Joke Bilcke, Mark Jit, Philippe Beutels.

**Methodology:** Marina Antillón, Lander Willem, Joke Bilcke, Mark Jit, Philippe Beutels.

**Project administration:** Joke Bilcke, Mark Jit, Philippe Beutels.

**Resources:** Joke Bilcke, Philippe Beutels.

**Software:** Marina Antillón, Lander Willem.

**Supervision:** Joke Bilcke, Mark Jit, Philippe Beutels.

**Validation:** Marina Antillón.

**Visualization:** Marina Antillón.

**Writing – original draft:** Marina Antillón, Xiao Li.

**Writing – review & editing:** Marina Antillón, Xiao Li, Lander Willem, Joke Bilcke, Mark Jit, Philippe Beutels.

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
