## [Editor Report · Decision Letter 0]

24 Jan 2023

Dear Dr Antillón, 

Thank you for submitting your manuscript entitled "The burden of respiratory syncytial virus in infants of low- and middle-income countries: A semi-parametric, meta-regression approach" for consideration by PLOS Medicine.

Your manuscript has now been evaluated by the PLOS Medicine editorial staff and I am writing to let you know that we would like to send your submission out for external peer review.

Please re-submit your manuscript within two working days, i.e. by Jan 26 2023 11:59PM.

Kind regards,

Callam Davidson

Associate Editor

PLOS Medicine

---

## [Decision Letter · Decision Letter 1]

9 Mar 2023

Dear Dr. Antillón,

Thank you very much for submitting your manuscript "The burden of respiratory syncytial virus in infants of low- and middle-income countries: A semi-parametric, meta-regression approach" (PMEDICINE-D-23-00070R1) for consideration at PLOS Medicine. 

[LINK]

In light of these reviews, I am afraid that we will not be able to accept the manuscript for publication in the journal in its current form, but we would like to consider a revised version that addresses the reviewers' and editors' comments. Obviously we cannot make any decision about publication until we have seen the revised manuscript and your response, and we plan to seek re-review by one or more of the reviewers. 

We hope to receive your revised manuscript by Mar 30 2023 11:59PM. Please email us (plosmedicine@plos.org) if you have any questions or concerns.

We look forward to receiving your revised manuscript. 

Sincerely,

Callam Davidson, 

PLOS Medicine

plosmedicine.org

Comments from the Academic Editor:

Please rewrite the text a bit to make it less technical and add a bit more interpretation of the results.

I found it a bit confusing that in the methods section there was a section entitled "Window of protection" and in the results section "Window of infection". Is that the same? Also, term window seemed a bit misleading, as it is about certain aspects of a distribution. I think the authors could do more to clarify what they mean by window of protection and how the results can be interpreted in that way.

One of the reviewers remarked on that the estimates were for the year 2020 and that the authors should comment on the impact of the pandemic. It was not clear for me that the estimates were meant to be for 2020, I thought that the authors implicitly assumed a sort of endemic (albeit seasonal) state. But I agree that some comments on the impact of the pandemic would be important.

Another reviewer suggested to not use two ways to calculate outcomes, but I found this actually good, because it gives additional validity to the estimates.

Requests from the Editors:

Please ensure that all numbers presented in the abstract are present and identical to numbers presented in the main manuscript text.

In the last sentence of the Abstract Methods and Findings section, please describe the main limitation(s) of the study's methodology.

Please remove the Code Availability, Role of the funding source, and Author Contributions sections from the main text and ensure the relevant information is captured in the Submission Form questionnaire.

For internet-derived references, please include the date accessed.

S1 Supporting Text, Figure A: Please confirm that the appropriate usage rights apply to the use of this map. Please see our guidelines for map images: https://journals.plos.org/plosmedicine/s/figures#loc-maps

Comments from the reviewers:

Reviewer #1: Antillon and colleagues reported a secondary analysis of the burden of RSV infections in infants of LMICs based on two global systematic reviews published in 2017 and 2022. The authors leveraged flexible generalized additive mixed models (GAMM) to estimate the age-dependent incidence and mortality of RSV infections, which would allow more flexibility for understanding the age distribution of RSV infections of different severity. The authors went ahead and estimated the global number of RSV infections and deaths in 2020, and compared those estimates with the two global systematic reviews. The authors showed that depending on the study settings, the peak, mean and median age of RSV infections could vary; these findings could help optimize the timing (of age) for RSV immunisation products given the anticipated short duration of protection. 

Overall, this analysis has its merits for an improved understanding of the age distribution of RSV cases and potentially deaths. The methodology for estimating the age-specific incidence and probability appears to be sound in general. However, the second part that estimated the absolute number of infections and deaths was problematic (see the comments below), and reading through the manuscript, it looks more like a validation of the model rather than a formal estimate; it is not essential to the main message of the manuscript. 

Comments for Part 1 - age distribution

* The authors assumed in their base model that the probability that a case in the community becomes a case in the hospital and that the hospitalized case becomes a fatality is itself age-dependent. This holds partly true - the probability is also expected to be dependent on country's income. 

o Accessibility and availability of hospital care is likely to be different across income groups (e.g., UMICs better than the other two), so is quality of health-care that is associated with in-hospital CFR. Although the authors explored income group as a fix effect in secondary model (and tested the added value albeit not being able to rule out false negative), I would consider swapping the two models and making income-specific splines the primary model. This is important especially if the age distribution of RSV cases and deaths differed by income (as seen in RSV GOLD studies). 

o I appreciate that the authors might have limited data from certain income regions (e.g., LIC) but that should not be the sole reason for not going for income-specific model as the primary model. If necessary, LIC and LMIC can be combined.

* Deaths in the community - I would leave out the deaths in the community (or move to the appendix at least) for two reasons. First, as the authors pointed out, data are sparse (compared to incidence and hospitalisation data) and are not suitable for GAMM models. Second, the alternative approach (that applied an inflation factor, Shi et al.) was based on an additional assumption about the proportion of RSV episodes that could access hospital care. The resulted estimates would not be as robust as the estimates of incidence and hospitalisations and undermined the overall quality of the work. 

Comments for Part 2 - estimation of absolute number

* I am confused by the decision to estimate the absolute number of global and regional RSV infections in the year of 2020 - the COVID-19 pandemic year when RSV epidemiology changed substantially (e.g., incidence, severity and age distribution). I do not see any additional efforts for accounting for the impact of the pandemic nor any justifications. 

o Reading through the manuscript, this part looks more like an informal validation of the GAMM models used above - then I would use the years of 2015 and 2019, respectively for validating the estimates by Shi et al and Li et al. This part could potentially go to the appendix.

Comments for "window of protection"

* I believe this is the meat of the paper and deserves more attention and work. Can the authors consider estimating the proportion of the overall burden in <1y or <5y for different RSV prophylactic products and for different schedules (e.g., birth dose or later in life)? This provides more straightforward practical information for decision-makers in addition to peak, mean and median age.

Other comments

* Writing style - while I appreciate that the authors described their methodology in great details, the overall text looks a bit too technical, especially considering the readerships of PLoS Med. This is for both the method and discussion sections. I spent almost half a day each to navigate through the amount of technical details.

o In the discussion section, I saw large amount of texts that compared numbers but few interpretation and appraisal of the estimates. 

Reviewer #2: This study addresses an important research question about the age profile of RSV disease burden in developing countries. It has relevance for understanding the potential impact of various RSV vaccination strategies. Overall, the study is well-done and clearly written. The manuscript is lengthy and presents multiple results. However, this manuscript can be better summarized and emphasize on the most important findings. Specific comments are as follows: 

Major comments:

- Lack of a clear description of the methods for different outcome models. Figure 1 panel A is very condensed, but the figure legend is more informative enough. The method description on page 6 is still not clear why the authors chose two different outcome models. I recommend the authors to phrase this part, including the following two burden models, as a base model and sensitivity analyses. 

- The section window of protection and the burden stratified by age in results should be combined and emphasized. This is the most significant and innovative finding this paper compared with others. Instead of presenting point estimates like mean, median, and peak age of infection, the authors should consider describing the distribution of the age profile and the impact of different vaccination strategies. Many previous papers already concluded the mean and median age of infection. In my opinion, what the author wrote in page 14 to 15, "…about a quarter to a half of all hospitalizations happen in the first 6 months of life…", is the most important finding for this manuscript. The author should consider expanding this part and discussing what percentage of children at risk can be protected with different type of vaccination strategy.

Minor comments:

- The current title does not reflect the content of the manuscript. I would suggest changing the title to "The detailed age distribution of RSV burden in infants of low- and middle-income countries: a generalized additive mixed effect model approach"

- The background of the abstract is disjoint from the following abstract. I would suggest adding a sentence to explain why estimate the age profile of RSV burden will help optimize the upcoming vaccination strategies.

- The probability of hospitalization is a conditional probability. The author should make it clear in the first place. It should be "the probability of hospitalization given infection".

- Page 2 line 17, statement: "maternal vaccines protect children no more than four months". Pfizer announced the results of a phase 3 clinical trial on maternal RSV vaccine and the protection is beyond 150 days.

- Page 2 line 25: The word "temporal resolution" is confusing. This paper does not measure RSV burden over calendar time as a time-series analysis but provide the detailed age profile of RSV burden.

- Page 4 line 81: The author should clarify that the "month-specific epidemiology" is age in month in pediatric populations.

- Page 8: "The scenario analysis: severity of RSV' seems to be disjoint from the main manuscript. The author should consider moving this section into the supplementary document and acknowledging the lack of severity measures as a limitation.

- Page 9 line 245 is unclear. What is the meaning of "the trend overall is higher"?

- Page 9 line 254. Large "amount".

- Page 9 line 259. What is the peak incidence? Hospitalization incidence? Community-based incidence?

- Page 9 line 263. For "12,77%", the decimal symbol should use the same format across the paper. 

Reviewer #3: Thanks for the opportunity to review your manuscript. My role is as a statistical reviewer, so my review concentrates on the study design, data, and analysis that are presented. I have put general questions first, followed by queries relevant to a specific section of the manuscript (with a page/line reference).

This manuscript estimates incidence of RSV (community, hospitalisations, and deaths from) across early childhood across a selection of lower and middle-income countries. Suitable data (with sufficient numbers of age-groups and from 200 onwards) were taken from a previous systematic review. An interesting approaching using generalised additive mixed models was applied - this enabled a country specific estimate of age-specific incidence of RSV with a random parameter for the spline effect of age. Sub-group effects were considered in the incidence and probability parts of the model, e.g. heterogeneity by country income group. Aggregated age-group data was dealt with my considering the total incidence/probability for an age-group to be representative of the mid-point of the interval. That is probably a reasonable approach for population level data. Shrinkage was employed to find an optimal number of knots for each of the splines. Two approaches to estimating burden were considered - these give very similar point estimates with OM generally having wider CI than OM I. This is a fairly complex approach - the detailed appendices (and github repository) were a great help for this review. The explanation and figures are well presented in terms of explaining the approach. The GAMMs used are a good way to estimate the target epidemiological model (e.g. incidence in community->hosp->death), but whether this model of burden is reasonable representation of the process, or if the underlying data from the systematic review is of sufficient quality to be used in this way is something I don't have the expertise to judge myself.

 I have worked with population level RSV data (a long time ago) and at the time I was surprised by the discrepancy between the burdens it represented in my own country compared to the relative level of attention paid towards this disease. I think that the overall aims and research questions proposed here are good (albeit this judgement comes from someone without substantial subject-matter expertise on RSV). 

One query I had was around generalisability of the findings from the included countries to all low and middle-income countries, e.g. good amounts of coverage for Central America, but limited data available for North Africa and Central Asia. The results should be internally valid to the countries included in the data, but I wonder if some of the language that generalises the results to all LMICs might not be appropriate?

Abstract, Methods and findings. I would reword the description of the secondary analysis so that it's clear that total RSV burden was estimated, at the moment the difference between the primary and secondary analyses is hard to distinguish because the phrases 'across different settings 'and 'across settings' look similar but actually mean completely different things. 

S.1.4 For the incidence part of the model, was over dispersion checked? Were residuals also examined for excess 0's? Given population is an offset I would guess this would unlikely to be an issue

S1.4.1 "…we excluded studies that fewer than three studies…" - should the last 'studies' here be 'age-groups'? 

S.1.4.2 What is the advantage of 5,000 draws from the MVN distribution compared to taking predictions directly from 5 and 95th percentile (or whatever limits are appropriate)?

[LINK]

---

## [Decision Letter · Decision Letter 2]

16 May 2023

Dear Dr. Antillón,

Thank you very much for re-submitting your manuscript "The age profile of respiratory syncytial virus burden in pre-school children of low- and middle-income countries: A semi-parametric, meta-regression approach" (PMEDICINE-D-23-00070R2) for review by PLOS Medicine.

I have discussed the paper with my colleagues and the academic editor and it was also seen again by 2 reviewers. I am pleased to say that provided the remaining editorial and production issues are dealt with we are planning to accept the paper for publication in the journal.

[LINK]

We look forward to receiving the revised manuscript by May 23 2023 11:59PM.   

Sincerely,

Philippa Dodd, MBBS MRCP PhD

PLOS Medicine

plosmedicine.org

Requests from Editors:

GENERAL

Thank you for detailed and considered responses to previous editor and reviewer comments. Please see below for further comments which we require that you address prior to publication.

Please start your numbering at line 1 of the abstract (as opposed to the introduction).

ABSTRACT

Final paragraph of methods and findings and the beginning of the conclusions, please temper the language used when reporting your results, the phrase, ‘we estimate...’ or similar may be helpful. 

AUTHOR SUMMARY

Thank you for including an author summary. The author summary should consist of 2-3 succinct bullet points under each of the following headings:

• Why Was This Study Done? Authors should reflect on what was known about the topic before the research was published and why the research was needed.

• What Did the Researchers Do and Find? Authors should briefly describe the study design that was used and the study’s major findings. Do include the headline numbers from the study, such as the sample size and key findings. 

• What Do These Findings Mean? Authors should reflect on the new knowledge generated by the research and the implications for practice, research, policy, or public health. Authors should also consider how the interpretation of the study’s findings may be affected by the study limitations. In the final bullet point of ‘What Do These Findings Mean?’, please describe the main limitations of the study in non-technical language.

We encourage you to review published articles on our website for examples. Please also see our author guidelines for more information: https://journals.plos.org/plosmedicine/s/revising-your-manuscript#loc-author-summary

STATISTICAL REPORTING

Page 9 onwards - when reporting 95% CIs, we suggest the use of commas to separate upper and lower bounds instead of hyphens as these can be confused with the reporting of negative values. Please check and amend throughout including all sections of the main manuscript text, figures (and tables) and supplementary files where relevant.

FIGURES

Please consider using/confirm usage of a colour palette suitable to those with colour blindness (i.e. avoiding red and/or green) to improve accessibility of your figures.

DISCUSSION

Please revise the structure of your discussion which should begin with a short, clear summary of the article's findings; what the study adds to existing research and where and why the results may differ from previous research; strengths and limitations of the study; implications and next steps for research, clinical practice, and/or public policy; one-paragraph conclusion. Please do not include any sub-headings in the text such that your discussion reads as a single piece of continuous prose. 

REFERENCES

For in-text reference callouts please remove spaces between citations, for example, line 24 (introduction) ‘…[20, 28, 29]…’ should read ‘…[20,28,29]…’ Please check and amend throughout all sections of the manuscript.

SUPPORTING INFORMATION

S1.4.1 and S1.8 – contains red text which may not be accessible to those with colour blindness. Please consider revising the use of red (and/or green).

S1 page 19 – please ensure that referencing format follows that of PLOS Medicine’s guidance which can be found here https://journals.plos.org/plosmedicine/s/submission-guidelines#loc-references

Please ensure that journal name abbreviations are those found in the National Center for Biotechnology Information (NCBI) databases. 

Please ensure that up to but no more than 6 author names are listed followed by et al, if more than 6 authors contribute to an individual study.

Please ensure that all web references include an access date.

SOCIAL MEDIA

If not already done so, to help us extend the reach of your research, please detail any Twitter handles you wish to be included when we tweet this paper (including your own, your co-authors’, your institution, funder, or lab) in the manuscript submission form when you re-submit the manuscript.

Comments from Reviewers:

Reviewer #1: No further comments.

Reviewer #3: Thanks for the revised manuscript and responses to my original queries. The updates and clarifications cover my original review - no further questions from me.

[LINK]

---

## [Editor Report · Decision Letter 3]

30 May 2023

Dear Dr Antillón, 

On behalf of my colleagues and the Academic Editor, Professor Mirjam Kretzschmar, I am pleased to inform you that we have agreed to publish your manuscript "The age profile of respiratory syncytial virus burden in pre-school children of low- and middle-income countries: A semi-parametric, meta-regression approach" (PMEDICINE-D-23-00070R3) in PLOS Medicine.

Prior to publication please ensure that you make the following revisions:

1) Author Summary

i) Line 53 – please split this statement as follows, ‘This study uses improved statistical models to estimate in 

depth the age profile of RSV cases, hospitalizations, and in-hospital deaths in young children.’

ii) If you wish to include the latter half of the statement, suggest the following – ‘This study may enable modellers 

to make improved estimates thus allowing policymakers to gain a better understanding of the potential impact 

that new pharmaceutical products could have.’ and placing as a final bullet point under ‘what do these findings 

mean’.

2) Discussion

Line 474 (and onwards) please change ‘is’ to ‘was’ or ‘was estimated at…’

PRESS

Best wishes,

Pippa 

Philippa Dodd, MBBS MRCP PhD 

PLOS Medicine